# Substitution of Cement with Granulated Blast Furnace Slag in Cemented Paste Backfill: Evaluation of Technical and Chemical Properties

**Soili Solismaa** [1,*]**, Akseli Torppa** [2]**, Jukka Kuva** [3]**, Pasi Heikkilä** [3]**, Simo Hyvönen** [4]**, Petri Juntunen** [4]**, Mostafa Benzaazoua** [5] **and Tommi Kauppila** [1]

1   Circular Economy Solutions, Geological Survey of Finland, 70211 Kuopio, Finland; tommi.kauppila@gtk.fi
2   Circular Economy Solutions, Geological Survey of Finland, 83500 Outokumpu, Finland; akseli.torppa@gtk.fi
3   Circular Economy Solutions, Geological Survey of Finland, 02150 Espoo, Finland; jukka.kuva@gtk.fi (J.K.); pasi.heikkila@gtk.fi (P.H.)
4   Department of Environmental Engineering, Savonia University of Applied Sciences, 70201 Kuopio, Finland; simo.hyvonen@savonia.fi (S.H.); petri.juntunen@savonia.fi (P.J.)
5   Mining Environment and Circular Economy, Mohammed VI Polytechnic University, Ben Guerir 43150, Morocco; mostafa.benzaazoua@uqat.ca
*   Correspondence: soili.solismaa@gtk.fi

**Abstract:** Cemented paste backfill (CPB) offers an environmentally sustainable way to utilize mine tailings, one of the largest waste streams in the world. CPB is a support and filler material used in underground mine cavities, which consists of mine tailings, water, and binder material that usually is cement. Replacing cement with secondary raw materials like granulated blast-furnace slag reduces the total $CO_2$ emissions and strengthens the internal microstructure of the CPB. This study characterizes the total- and soluble contents of CPB starting materials and five CPB specimens containing different levels of slag substitution. In addition, phase composition (mineral liberation analysis, MLA) and internal structure (X-ray tomography) of five CPB specimens is documented, and measurements of compressive strength are used to evaluate their suitability as backfill material. Mine tailings and CPB specimens used in this study are rich in sulphates and arsenic, but low in sulphides. Stronger As leaching of ground CPB specimens compared with ground mine tailings is related to the elevating pore water pH during the cement hydration. The hydration product ettringite is found in all CPB specimens and its content is the lowest in the slag containing specimens. X-ray tomography revealed vertically differentiated density structures in the CPB specimens. The lower parts of all specimens are denser in comparison with the upper parts, which is probably due to the compaction of the solid particles at the base. The compressive strength test results indicate that partial substitution of cement with slag improves the strength of the CPB. The total replacement of cement with slag reduces the early strength but gives excellent strength and lower porosity over longer time intervals. The results of the study can be utilized in developing more durable and environmentally responsible CPB recipes for gold mines of similar mineral composition and gold extraction method.

**Keywords:** cemented paste backfill; mine tailings; cement; granulated blast furnace slag; mining environments; circular economy

## 1. Introduction

Extractive waste is one of the largest waste streams in the world, consisting 26% of all waste in the EU [1]. It consists mainly of waste rocks and mine tailings. The demand for mineral products is predicted to increase by 500% by 2050 [2]. This will dramatically increase the amount of extractive wastes if they are not utilized more effectively. The mining of sulphide rich ores may result in an environmental problem called acid mine drainage, AMD. For example, when mine tailings rich of sulphide minerals are exposed to water and oxygen in tailings impoundments, the sulphide minerals are oxidized to form

acidic sulphate ($SO_4^{2-}$) and metal rich seepage waters [3,4]. In addition, non-sulphidic extractive wastes are also capable of causing contamination of ground and surface waters, in which they release contaminants such as sulphates, metalloids, or metals.

One method of reducing the disadvantages caused by mine tailings is stabilization and utilization of mine tailings as a support and filler material in underground mine facilities. The use of cemented paste backfill (CPB) reduces the volume of surface disposed mine tailings, which decreases the operating costs and the environmental impacts of the mine. Acid mine drainage (AMD) can be decreased when tailings are mixed with binders, which provides extra neutralization potential, and are stored in less oxidizing conditions underground [5]. In addition, the use of CPB in underground mines enables more efficient extraction of the ore, as fewer supporting pillars are needed [6–8].

CPB consists of mine tailings, water and a binder material that is usually cement. Cement production causes $CO_2$ emissions e.g., due to the calcination of calcium carbonate ($CaCO_3$). Substituting part of the cement with secondary raw material, granulated blast-furnace slag (slag), in CPB production could thus reduce the carbon footprint of the mine. Another advantage of slag is that it improves the quality of the cement-based materials by creating lower permeability and reducing the likelihood of sulphate attack. Permeability and porosity diminish when slag reacts with hydrated lime to form extra C–S–H gel, which is the main binding phase in hardened cement [9]. The addition of slag also decreases delayed ettringite formation, which might cause expansion or cracking in cement-based material in later phases [9]. This is caused by the additional $Al_2O_3$ content of slag decreasing the $SO_3/Al_2O_3$ ratio [10] and hydraulic pozzolan reactions of slag which lead to the formation of additional hydrates [11]. These properties make slag a suitable binder when sulphates are present in the parent materials [6,12,13].

When a mine is in operation, CPB is typically placed in dry unflooded cavities where it is exposed to varying degrees of contact with seepage waters and oxygen. Final closure of a mine typically results in flooding of the underground mine and the CPB, markedly decreasing the oxidation of sulphide minerals [14]. Mechanical and chemical properties of backfill material should be studied at an early stage, as chemical alteration of CPB in the presence of mine waters can seriously affect the durability of the backfill material.

Neutral and slightly alkaline conditions might prevent AMD and the detrimental effects caused by it. Most metals are stable in alkaline conditions, but elements such as As are mobilized in neutral and alkaline conditions [15]. The pH rise during the hydration of cement can result in leaching and re-crystallization of As. Hamberg et al. [16] concluded that the addition of binders to tailings increased As leaching due to the relocation of desorbed As from As-bearing Fe hydrates into less acid-tolerant species such as Ca-arsenates and cementitious As-phases. Benzaazoua et al. [5] state that the addition of hydraulic Ca-rich binders enhances the precipitation of calcium arsenates or sometimes Ca-arsenites. Ca-arsenates are stable at pH > 10 but become more soluble at lower pH values.

CPB reactivity depends on the properties and ratio of tailings, water and binders used in CPB preparation. In this study, we compare the leaching test results of five CPB test specimens to the Finnish threshold values to evaluate the environmental properties of different CPB materials. Our aim is to obtain a compressive strength of 0.5 MPa after 14 days of curing, which is required by the mine. Detailed aims of the study are to prepare CPB test specimens using different recipes and:

a. Study the leaching behavior and changes in phase composition of the five different recipes;
b. Test if the compressive strength of the alternative CPB recipes meets the structural requirements and how the different binder mixtures affect attainment of CPB strength;
c. Study how the changes in the recipes affect the internal microstructure of the specimens.

## 2. Materials and Methods

In this study, the chemical properties of CPB parent materials (cement, slag, mine tailings, pore water and process water) and five CPB recipes were investigated. The

differences in chemical and phase composition in addition to the internal structure of the CPB test pieces were studied to find out how the replacement of cement by slag affects these properties. The mine tailings used in this study were very low in sulphides but abundant in sulphates, arsenic and metals. The process water used in CPB preparation originated from the internal treatment process of the mine and was also rich in sulphates, As, and metals.

The parent material properties were first characterized for their phase composition and chemical properties. Several parallel test pieces were prepared to examine their compressive strengths, chemical and phase compositions, and 3D structures. Methods of characterizing the parent materials, preparing the CPB test pieces, characterizing the test pieces for phase and chemical composition, measuring compressive strength, and performing X-ray tomography investigation are described in this Section 2.

### 2.1. Materials

The materials used for CPB preparation consists of fresh mine tailings, process water, Portland cement (GU-type), and granulated blast-furnace slag.

Wet mine tailings and process water samples for the preparation of CPB specimens of the Kittilä gold mine in Finland were sampled. The host rocks of the deposit are ~2.02 Ga orogenic mafic volcanic rocks [17]. Mineralized rocks are hydrothermally altered, and the gold occurs in arsenopyrite and pyrite [17]. Once extracted, the ore enrichment process includes crushing, grinding, and a bulk flotation process, which produces a gold-rich sulphide concentrate that is cyanided for gold extraction and leaves behind wet low-sulphide mine tailings [18]. The latter is used to produce CPB at the mine and is also used and compared in this study. Process water is used in the flotation processes and treated with $Ca(OH)_2$ before utilized in CPB preparation.

The slag and cement were obtained from Finnsementti Ltd (Parainen, Finland). The slag product used in CPB test samples was granulated blast-furnace slag KJ400 that is formed as a side product of the iron smelting process, when the alkaline melt is cooled rapidly. The cement is called Plussementti CEM II/B-M (S-LL) 42.5 N that itself contains 10–25% of blast furnace slag KJ400.

### 2.2. Methods

#### 2.2.1. Process Water and Mine Tailings Porewater

A characterization of the properties of water used in CPB preparation is essential for the evaluation of the chemical, physical, and phase changes that occur in the CPB mixtures. The currently used CPB at the mine contains tailings, pore water and process water. Process water from the mine was sampled for chemical analyses and used in making the CPB specimens. Tailings pore water samples were taken from the water layer that was segregated on top of the tailings after transport.

In the Eurofins Labtium Ltd laboratory (Espoo, Finland) the samples were filtered using a 0.45 μm filter and preserved with nitric acid ($HNO_3$) before measurements. The determinations of 33 dissolved elements were performed by inductively coupled plasma mass spectrometry (ICP-MS) and inductively coupled plasma optical emission spectrometry (ICP-OES) [19]. The dissolved anions Br, Cl, F, NO3 and $SO_4^{2-}$ were determined by liquid chromatography for ions [20] from samples without the nitric acid preservative.

#### 2.2.2. Mine Tailings, Slag, Cement and Five Cemented Paste Backfill (CPB) Recipes

The sample treatments and measurements of elements were made in the accredited laboratory of Eurofins Labtium Ltd (Kuopio, Finland) [21]. The solid materials used for CPB preparation were lime-washed wet mine tailings, cement, and slag. The solid parent material samples were freeze-dried and ground in a hardened steel container for the chemical leaching analyses. The (near) total concentrations of elements in the tailings, slag and cement samples were determined using a four-acid leaching method. Leaching with aqua regia was used to determine the acid extractable element concentrations, which

is often a good indicator of metal release in AMD conditions [22]. Aqua regia digests inter alia trioctahedral micas, clay minerals, sulphides, saline minerals, and secondary precipitates [23,24]. The concentrations of cations, anions and elements in the leachate were measured by inductively coupled plasma mass spectrometry (ICP-MS) and inductively coupled plasma optical emission spectrometry (ICP-OES). The two-stage batch test is a two-stage leaching process at liquid–solid ratios of 2:1 and 8:1 [25]. For this method, the samples were sieved to <2 mm grain size. The cumulative content of solid materials was calculated according to the standard. The CPB specimens (see below) were crushed prior to grinding for the leaching tests but otherwise analyzed according to a similar protocol to the parent materials.

For the mineralogical characterization of solid parent materials, vertical polished sections were prepared at the Geological Survey of Finland (GTK), Outokumpu Mintek from the air-dried tailings, slag, and cement samples. The specimens were first mounted in epoxy, the cured epoxy mounts were cut into vertical slices, the slices were rotated 90 degrees, mounted again in epoxy and polished. This technique is particularly suitable for granular materials as it minimizes the effects of settling of dense or coarse mineral particles during the specimen preparation stage. Conventional polished thick sections were prepared from the middle of each solid CPB specimen.

The characterization included the identification and quantification of mineral phases using an FEI Quanta 650 FEG-SEM with two Bruker X-flash energy-dispersive X-ray (EDX) detectors equipped with Mineral Liberation Analyzer (MLA) software. The measurement parameters were an acceleration voltage 25 kV, vacuum <5.5 × $10^{-6}$ mbar. The MLA measurement technique is described in detail by Sylvester [26]. Mineral identifications were based on Bruker's mineral database. However, the database does not contain data for typical cement hydration phases. Mine tailings grain size distribution was measured with the laser diffraction method using a Malvern Mastersizer 2000.

### 2.2.3. Preparation of CPB Test Pieces

Five CPB recipes were designed based on the current one used in the operating gold mine. The objective was to find a suitable amount of slag to replace cement without sacrificing environmental or technical and structural properties of the CPBs. The recipe currently used at the mine (R4 100:0; Table 1) contains only cement as a binder. The cement currently used (CEM II/B-M (S-LL) 42.5 N) at the mine already contains 10% to 25% slag and the same cement was also used in the modified recipes. The final amount of water used was 30% for each recipe. Water consisted of mine tailings pore water and 4% of process water (except for R5) that was added to adjust the water content of the mixtures. For the actual CPB that is currently in operation at the mine, the lime-washed mine tailings are first dried in the paste plant after which process water is added to meet the planned 30% water content. In contrast, the tailings were not dried for the present study and most or all (R5) of the water in the CPB mixtures was porewater, which was present in the tailings. The properties of these two waters are quite similar and presented in Section 3.1.1.

The main component of the test pieces was gold mine tailings (93–95% of the dry weight). Cement and slag were used as a binder material at different proportions (cement:slag) thus, R1 40:60, R2 and R5 50:50, R3 0:100, and R4 100:0, respectively. The proportion of binder was 7% and a 5% version R5 was made of the 50:50 recipe.

At Savonia University of Applied Sciences a 1 kg sample of tailings was dried in an oven (105 °C ± 5 °C) and weighed before and after heating to determine the dry weight of the tailings and to calculate the amount of binder and the water to be added to the material mixtures. Fresh mine tailings and process water (when needed) were added first to the mixing vessel. A Scheppach PM1800D mortar mixer was used to mix the paste backfill mixtures. Mine tailings and water were blended for 15 min, followed by the addition of binder materials, and mixing for another 10 min, or until the mixture seemed homogenous.

**Table 1.** The percentages of used materials for the five studied recipes R1–R5.

| Starting Materials | Cemented Paste Backfill Specimens (Cement: Slag Ratio) | | | | |
|---|---|---|---|---|---|
| | R1 (40:60) | R2 (50:50) | R3 (0:100) | R4 (100:0) | R5 (50:50) |
| Mine tailings (%) | 63 | 63 | 63 | 63 | 65 |
| Water tot. (%) | 30 | 30 | 30 | 30 | 30 |
| Share of pore water (%) | 96 | 96 | 96 | 96 | 100 |
| Share of process water (%) | 4 | 4 | 4 | 4 | 0 |
| Binder (%) | 7 | 7 | 7 | 7 | 5 |
| Share of cement from binder (%) | 40 | 50 | 0 | 100 | 50 |
| Share of slag from binder (%) | 60 | 50 | 100 | 0 | 50 |
| Consistency slump mm | 200 | 220 | 210 | 190 | 200 |

The Envitop Oy plastic 77 mm × 150 mm molds were treated with form oil (Sem®Form BE) for the compression strength determination. Filling the molds was performed in two stages; first the mold was half filled with the mixture, struck against the table until the largest air bubbles were eliminated, another layer was filled on top of the mold, and air bubbles were again eliminated as described above. More material was added until the mold was full, and it was finished with a spatula.

The filled molds were left to set for an hour without allowing any drainage, after which they were sealed with a cap and placed in a climate test chamber (Vötsch VC4034). The conditions simulated underground conditions with a temperature of 16 °C and a humidity of 98%.

### 2.2.4. Compressive Strength Tests of the CBP Blocks

Compressive tests are an important indicator of the technical properties of the different CPB recipes. Low strength of the material might mean structural instability and a predisposition to release harmful substances in the long term. Unconfined compressive strength of the CPB specimens was tested at Savonia University of Applied Sciences to study and ascertain whether they met the technical requirements at the mine: at least 0.5 MPa achieved by 14 days of curing.

Compressive strength tests were performed according to the standards [27,28] in an accredited concrete testing laboratory [29]. Test pieces were unwrapped from the molds before compressive strength tests by opening the side of the mold. If the upper surface was detected to be irregular, it was cut with a clipper diamond saw.

The diameter of the cylindrical specimens was measured at six points (upper and lower surface) and the height at three points, after which the specimen was weighed in order to determine the density needed for the calculation of the final compressive strength.

Compressive strength was measured using a Controls Automax Compact Pro-press that is designed for compression measurement of concrete. The approach speed of the press plate was reduced for CPB, and the load speed was set to 0.25 MPa/s. The lowest calibration point of the press is 20 kN. If the load force falls below this, the result is not reliable. Therefore, comparison tests were performed by Envitop Ltd, using a device designed for lower loads, and the results were congruent with the compressive strength tests performed by Savonia University of Applied Sciences.

### 2.2.5. X-ray Tomography

The CPB specimen's internal structures were evaluated using the X-ray computed tomography (XCT) method. The samples were scanned with a GE phoenix v|tome|x s tomography device at GTK Espoo research laboratory to study the effects of slag substitution on the internal structure of the CPB specimens. The imaging was undertaken through the plastic liner. A 240 kV microfocus tube was used with a 120 kV accelerating voltage and a 1.075 mA tube current for a total power of 129 W. 0.5 mm of Cu was used as a beam filter and a voxel resolution of 88.16 μm was achieved. At 1500 angle steps, the detector first paused for a single exposure time and then took an average of over three exposures.

A single exposure time was 200 ms for a total scan time of 20 min per sample. No ring artefact reduction was undertaken, and the beam hardening correction coefficient used in reconstruction (0–10) was 6.

After reconstruction, the images were converted to 16-bit depth (from 32-bit) and their overall volume was determined using 2-phase watershed segmentation in ThermoFisher PerGeos 2020.2 (Waltham, MA, USA). The pore space within the sample was similarly segmented with a two-phase watershed and the pores finally separated with the Labeling tool. This allowed the measurement of the porosity and number of pores in each sample.

## 3. Results and Discussion

### 3.1. Chemical Composition of Parent Materials and the CPB Recipes

3.1.1. Process Water and Mine Tailings Porewater

The pore water of the mine tailings is high in sulphate, Mg and Ca, but also contains relatively high K and Na (Table 2). A very small amount of process water from the mine was added to the CPB test specimens, but it provided lower dilute sulphate and Mg content and higher Ca-content relative to the pore water. The pH of both was basic, and the process water provided higher alkalinity than pore water.

The ICP measurement results provide slightly lower S contents (12–14% relative) when compared to the $SO_4^{2-}$ analysis results. The high $SO_4^{2-}$ content of the pore water and process water might have had an activating effect on Ca rich slag.

3.1.2. Solid Materials

All the parent materials were rich in Ca, S, Al, Fe and Mg (Table 3). Compared to cement, the used slag was lower in Ca and S but notably higher in Al, Fe, Mg and Al, which should enhance the development of aluminate phases with a higher proportion of slag. The mine tailings (MT) were low in Ca in comparison with cement and slag.

R3 with only the slag binder had the lowest Ca and Fe and highest Mg concentrations of CPB specimens. R4 with only cement binder had the highest Ca concentration and the lowest Al concentrations of the CPB samples. Total concentrations of As and Fe, were highest in R5 where the cement–slag ratio was 50:50 and the binder amount low (5%). Slag and cement contained only trace amounts of As (1.3 and 22.4 mg/kg respectively) whereas the As content of the mine tailings was relatively high (1560 mg/kg), which was reflected in the relatively higher As content found in the test samples.

The aqua regia (AR) leaching test values indicated that using 5% and 7% inclusions of binder material would not be sufficient to chemically stabilize mine tailings, if the CPB were to be exposed to severe AMD. However, it would be improbable for this type of tailings to be the source of the AMD since the tailings were poor in sulphide-bearing minerals.

The AR leaching results suggested that As, Cu, and Ni would be the elements of greatest concern if the CPB material were to be subject to intensive weathering, and these were followed by Sb and Co. The parent tailings and all CPB specimens exceeded the threshold values, lover guideline values and upper guideline values announced in a Finnish management strategy for contaminated land [30] for several elements in the aqua regia leaches (Table 4). The values were determined by using the same leach test. Mine tailings exceeded upper guideline values for As, Cu and Ni. Sb exceeded its lower guideline value, and concentration of Co exceeded its threshold guideline value. Slag exceeded the upper guideline value for contaminated soils for vanadium whereas other concentrations of elements in the slag did not exceed any of the guideline values. Cement exceeded the threshold value for Sb, As, Co, Cu and V and the lower guideline value for Zn. CPB samples R1-R5 exceeded the limits in a similar manner as the mine tailings, although the values were slightly lower in these binder-containing samples. In contrast, the contents of V and Zn were low in the CPB specimens, despite the values being high in slag and cement.

**Table 2.** Properties and chemical contents of process water and tailings pore waters used in the preparation of the CPB mixtures.

| Measured Variable | Detection Limit | Unit | Pore Water | Process Water |
|---|---|---|---|---|
| Alkalinity | 0.1 | mmol/L | 1.4 | 4.7 |
| pH | - | pH | 8.0 | 7.8 |
| $SO_4^{2-}$ | 50 | mg/L | 7900.0 | 2300.0 |
| S | 1 | mg/L | 2260.0 | 676.0 |
| Mg | 0.05 | mg/L | 1410.0 | 103.0 |
| Ca | 0.1 | mg/L | 357.0 | 578.0 |
| K | 0.01 | mg/L | 114.0 | 79.0 |
| Na | 0.2 | mg/L | 101.0 | 117.0 |
| Cl | 10 | mg/L | 25.0 | 28.0 |
| F | 1 | µg/L | 1800.0 | <1.0 |
| Mn | 0.02 | µg/L | 1350.0 | 1440.0 |
| Sr | 0.1 | µg/L | 1180.0 | 1950.0 |
| Li | 0.1 | µg/L | 149.0 | 233.0 |
| Rb | 0.01 | µg/L | 134.0 | 138.0 |
| P | 0.05 | µg/L | 100.0 | 500.0 |
| As | 0.05 | µg/L | 58.9 | 17.5 |
| Mo | 0.02 | µg/L | 35.2 | 6.3 |
| Ba | 0.05 | µg/L | 20.7 | 55.8 |
| Al | 1 | µg/L | 17.7 | 13.2 |
| Ni | 0.05 | µg/L | 11.2 | 63.5 |
| Cu | 0.1 | µg/L | 10.6 | 3.8 |
| Co | 0.02 | µg/L | 2.7 | 6.1 |
| Zn | 0.2 | µg/L | 0.4 | 9.5 |
| V | 0.05 | µg/L | 0.1 | 0.4 |
| Cr | 0.2 | µg/L | <0.2 | 6.0 |
| Cd | 0.02 | µg/L | 0.0 | 0.1 |

**Table 3.** Results of four acid leaching methods (total concentrations). The results of CPB specimens are presented as a mean value of two samples.

| Element | Starting Materials | | | Cemented Paste Backfill Specimens (Cement:Slag Ratio) | | | | |
|---|---|---|---|---|---|---|---|---|
| | Cement | Slag | Tailings | R1 (40:60) | R2 (50:50) | R3 (0:100) | R4 (100:0) | R5 (50:50) |
| Al % | 3.1 | 6 | 5.1 | 4.4 | 4.4 | 4.4 | 4.2 | 4.4 |
| Fe % | 2.2 | 9.7 | 9 | 5.9 | 5.9 | 5.8 | 5.9 | 6.1 |
| Mg % | 2.3 | 7.2 | 2 | 3.2 | 3.2 | 3.3 | 3 | 3.2 |
| Ca % | 38.5 | 25.2 | 6.2 | 7.8 | 7.9 | 7.4 | 8.1 | 7.5 |
| Na % | 0.5 | 0.4 | 1.9 | 1 | 1 | 1 | 1 | 1 |
| K % | 0.8 | 0.7 | 1.1 | 1.2 | 1.2 | 1.2 | 1.2 | 1.2 |
| S % | 1.4 | 0.8 | 2 | 1.7 | 1.8 | 1.7 | 1.7 | 1.8 |
| As (ppm) | 22 | 1 | 1560 | 1570 | 1580 | 1550 | 1565 | 1640 |
| Ba (ppm) | 260 | 723 | 202 | 194 | 193 | 205 | 175 | 188 |
| Cd (ppm) | 0 | <0.1 | 0 | 0 | 0 | 0 | 0 | 0 |
| Co (ppm) | 38 | <2.0 | 29 | 34 | 35 | 33 | 36 | 35 |
| Cr (ppm) | 52 | 31 | 46 | 257 | 263 | 258 | 265 | 265 |
| Cu (ppm) | 136 | 3 | 265 | 259 | 264 | 255 | 263 | 267 |
| Mo (ppm) | 18 | <2.0 | <2.0 | <2.0 | <2.0 | <2.0 | 3 | <2.0 |
| Ni (ppm) | 58 | 3 | 50 | 158 | 161 | 156 | 159 | 167 |
| Pb (ppm) | 20 | <10.0 | <10 | 13 | <11.0 | 13 | 16 | 12 |
| Sb (ppm) | 4 | <0.2 | 14 | 21 | 21 | 21 | 21 | 20 |
| V (ppm) | 142 | 355 | 88 | 198 | 200 | 204 | 189 | 199 |
| Zn (ppm) | 325 | <2.0 | 97 | 96 | 101 | 95 | 107 | 99 |

**Table 4.** Comparison of five CPB recipes with aqua regia leaching results to guideline values of Finnish management strategy of contaminated land [30]. Explanation for R1–R5, see Table 1. D denotes duplicate sample.

| | Sb mg/kg | As mg/kg | Cd mg/kg | Co mg/kg | Cr mg/kg | Cu mg/kg | Pb mg/kg | Ni mg/kg | Zn mg/kg | V mg/kg | S % |
|---|---|---|---|---|---|---|---|---|---|---|---|
| Treshold value | 2 | 5 | 1 | 20 | 100 | 100 | 60 | 50 | 200 | 100 | |
| Lower guideline value | 10 | 50 | 10 | 100 | 200 | 150 | 200 | 100 | 250 | 150 | |
| Upper guideline value | 50 | 100 | 20 | 250 | 300 | 200 | 750 | 150 | 400 | 250 | |
| Blast-furnace slag | <0.02 | 1.15 | <0.01 | <1 | 29.6 | 2 | <0.1 | 2.2 | <1 | 321 | 0.63 |
| Cement | 3.65 | 18.3 | 0.22 | 27.4 | 60.8 | 117 | 20 | 48.3 | 287 | 102 | 0.12 |
| Mine tailings | 20.4 | 1730 | 0.26 | 35.1 | 78.4 | 288 | 3.85 | 164 | 81 | 42.8 | 1.81 |
| R1 (40:60) | 16.8 | 1500 | 0.24 | 33.9 | 85.9 | 260 | 4.57 | 157 | 89 | 63.5 | 1.73 |
| R1 (40:60), D | 17.2 | 1490 | 0.23 | 32.7 | 86.2 | 260 | 4.25 | 155 | 89 | 63.9 | 1.73 |
| R2 (50:50) | 19.4 | 1490 | 0.24 | 31.4 | 82.9 | 256 | 4.24 | 151 | 86 | 57.8 | 1.63 |
| R2 (50:50), D | 17 | 1510 | 0.23 | 33.5 | 82.9 | 259 | 4.29 | 155 | 89 | 60.3 | 1.73 |
| R3 (0:100) | 19.4 | 1510 | 0.23 | 32.6 | 83.7 | 261 | 3.61 | 157 | 81 | 70.2 | 1.75 |
| R3 (0:100), D | 17.9 | 1500 | 0.24 | 31.5 | 81.2 | 252 | 3.63 | 151 | 78 | 67 | 1.69 |
| R4 (100:0) | 20.4 | 1540 | 0.26 | 35.5 | 87.6 | 271 | 2.44 | 163 | 103 | 55.9 | 1.77 |
| R4 (100:0), D | 19.1 | 1510 | 0.24 | 34.2 | 86.8 | 271 | 2.37 | 161 | 102 | 54.6 | 1.76 |
| R5 (50:50 5%) | 19.3 | 1520 | 0.25 | 34.5 | 89.3 | 272 | 4.09 | 162 | 91 | 61.1 | 1.79 |
| R5 (50:50 5%), D | 18.6 | 1520 | 0.23 | 33.8 | 87 | 267 | 4.04 | 161 | 90 | 59.1 | 1.79 |

In addition, aqua regia leachable S content was high in mine tailings and all the CPB samples. The mineralogy results (Section 3.2) suggested that $SO_4^{2-}$ was bounded in gypsum ($CaSO_4 \cdot 2H_2O$). Gypsum can decay in acid conditions [31], and cause $SO_4^{2-}$ load for the mine waters.

A two-step batch leaching test [25] is normally used for landfill waste classification in Finland [32]. In this particular study it was also used to assess the solubility of elements of the CPB specimens even though the method is not intended for mine waste classification. Because the studied mine tailings are not acid generating, the two-step water leaching method was better suited to evaluate the leaching behavior of the CPB than the AR leaching test. Although the AR leaching tests showed little differences between the recipes, certain differences appeared when using the two-step batch leaching tests (Table 5). Mo, F and $SO_4^{2-}$ contents of cement exceeded their respective inert waste threshold values. The mine tailings exceeded inert waste threshold values for As and $SO_4^{2-}$.

**Table 5.** Two-step batch leaching test results comparing the parent material and CPB-recipes leachates against current threshold values as stipulated by the Finnish government degree on landfills [32]. Sb, Cd, Pb, Ni, Se, Zn results are below detection limit. Explanation for R1–R5, see Table 1. D denotes duplicate sample.

| | As mg/kg | Ba mg/kg | Cr mg/kg | Cu mg/kg | Mo mg/kg | F− mg/kg | Cl− mg/kg | $SO_4^{2-}$ mg/kg | DOC mg/kg |
|---|---|---|---|---|---|---|---|---|---|
| Inert waste | 0.5 | 20 | 0.5 | 2 | 0.5 | 10 | 800 | 1000 | 500 |
| Non-hazardous waste | 2 | 100 | 10 | 50 | 10 | 150 | 15,000 | 20,000 | 800 |
| Hazardous waste | 25 | 300 | 70 | 100 | 30 | 500 | 25,000 | 50,000 | 1000 |
| Blast-furnace slag | <0.09 | 1.2 | <0.1 | <0.05 | <0.05 | 1.5 | <1 | 129 | 8.2 |
| Cement | <0.05 | 16.6 | 0.4 | <0.05 | 6.9 | 18.4 | 176.6 | 2697 | 81.5 |
| Mine tailings | 0.6 | 0.1 | <0.1 | <0.05 | <0.05 | 8.4 | 15.8 | 19,470 | 5.9 |
| R1 (40:60) | 1.9 | 0.6 | <0.07 | 0.1 | 0.3 | 1.1 | 26.5 | 13,219 | 11.5 |
| R1 (40:60), D | 1.9 | 0.6 | <0.07 | 0.1 | 0.4 | 4.3 | 30.2 | 12,896 | 10.8 |
| R2 (50:50) | 1.6 | 0.6 | <0.06 | 0.1 | 0.5 | 4.8 | 31 | 13,175 | 12.4 |
| R2 (50:50), D | 1.7 | 0.6 | <0.06 | 0.1 | 0.4 | 4.6 | 31.9 | 13,613 | 11.9 |
| R3 (0:100) | 1.4 | 0.7 | <0.06 | <0.06 | 0.1 | 4.1 | 23.2 | 12,409 | 6.9 |
| R3 (0:100), D | 1.5 | 0.7 | <0.06 | <0.05 | 0.1 | 4.2 | 23.4 | 12,583 | 7.6 |
| R4 (100:0) | 1.7 | 0.5 | 0.1 | <0.06 | 0.7 | 6.6 | 40.6 | 12,321 | 12.9 |
| R4 (100:0), D | 1.6 | 0.5 | <0.05 | <0.05 | 0.8 | 6.5 | 40 | 12,825 | 12.6 |
| R5 (50:50) | 2.2 | 0.5 | 0.1 | <0.05 | 0.3 | 4.4 | 30.6 | 12,127 | 8.8 |
| R5 (50:50), D | 2.2 | 0.5 | 0.1 | <0.05 | 0.3 | 4.4 | 30.1 | 12,882 | 9.5 |

The binder addition failed to prevent the leaching of $SO_4^{2-}$ and appeared to increase the leaching of As in the samples when compared to the parent materials. A similar observation was made from cyanidation tailings of the Kittilä gold mine in an earlier study [16]. Both As and $SO_4^{2-}$ exceeded the reference concentrations of inert waste for all the CPB samples. An explanation for this may be that oxyanion-forming metalloids such as As are mobilized in circumneutral to slightly alkaline conditions in pH 6–9 [33], whereas most metals are less mobile in these conditions [34]. The pH elevation is a consequence of adding alkaline binder material to the tailings. Interestingly, Sb, another oxyanion-forming element that leached out in the AR extraction, was not detected in the water-based leaches of any of the samples. The waters that flush CPB fillings in underground mine cavities can have oxidizing properties during the mine operating period and the binder materials increase the water's alkalinity. After the final closure, when the mine fills with groundwater, the conditions typically become oxygen deficient.

Sample R5 (50:50) with the binder content of only 5% exceeded the As threshold value for non-hazardous waste in the water leaching tests. An explanation for the elevated As content in R5, in which the binder content was lower than in the other samples, might be that R5 contains a higher proportion of As containing mine tailings and pore water (see Tables 1 and 2) and the lower binder content may also have a pH effect on As leaching. R3 (0:100) showed a slightly lower As leachability, when compared with the other recipes. The major trace metals of concern in the tailings, Ni and Cu, did not exceed reference values for any of the CPB mixtures and neither did Sb and Co.

The results of the leaching tests cannot be directly upscaled to large CPB volumes underground. It is mostly the surface of the underground CPB body that is flushed by the percolating waters while most of the mass remains unreacted. Moreover, these samples were ground to less than 4 mm grain size for the batch leaching test, an action which increases the samples' reactive surface area. Neither the actual surface areas nor the exact grain size distribution of the test materials were measured in this study. Our aim, however, was to compare the different CPB recipes and to identify of the main contaminants of concern in the CPBs. In addition, our results represent a snapshot of the tailings material, pore waters, and process waters, the composition of which may vary over time, depending on the mill feed.

### 3.2. Mineral and Secondary Phase Composition of Parent Materials and the CPB Recipes

Phase composition (Table 6) of the starting materials (tailings, slag, cement) and the CPB test specimens (R1–R5) of polished sections were analyzed using a scanning electron microscope (SEM) equipped with a mineral liberation analyzer (MLA) to study the phase reactions in CPBs after 60 days of curing. Based on the MLA data, around 99 wt.% of the slag consisted of Ca-Mg-Al-silicate glass (slag), with Fe-oxide inclusions (Figure 1A). Accessory phases include dolomite, quartz and other silicates, and magnetite ($Fe_3O_4$). Cement was predominantly (>60%) composed of Ca-silicate, Ca-aluminate, and Ca-Al-Fe-oxide, which according to common cement nomenclature are defined as $C_2S$, $C_3S$, $C_3A$, and $C_4AF$, respectively. In addition, Ca-Mg-Al-silicate (slag), calcite, gypsum, lime (CaO), and quartz along with other minor silicates, and metal oxides ($Fe_2O_3$, $Al_2O_3$, MgO), were observed in the cement. The major mineral components of the mine tailings (Figure 1B) were quartz (20 wt.%), feldspars (21 wt.%), carbonates (21 wt.%), mica and chlorite (11 wt.%), metal oxides and hydroxides (10 wt.%) and gypsum (10 wt.%). Regarding sulphide minerals, only pyrite (0.2 wt.%) and arsenopyrite (0.2 wt.%) were detected. The MLA analyses revealed feldspars that were predominantly albite, mica was mostly muscovite, Ca-sulphate gypsum, and carbonates represented Mg-Fe-rich variants that belong to a dolomite-ankerite series. The metal oxides and hydroxides mainly comprised oxides and hydroxides of iron.

**Table 6.** Measured mineral and phase compositions of tailings, slag, cement, and CPB specimens (R1–R5). Cement to slag ratios used in CPB preparation are given along with the specimen codes. "–" = not detected.

| Minerals and Phases (wt.%) | Starting Materials | | | CPB Samples (Cement: Slag) | | | | |
|---|---|---|---|---|---|---|---|---|
| | Tailings | Slag | Cement | R1 (40:60) | R2 (50:50) | R3 (0:100) | R4 (100:0) | R5 (50:50) |
| Quartz | 20 | - | 0.3 | 21.3 | 24.4 | 21.9 | 21.3 | 22.8 |
| Feldspars | 21.1 | - | - | 19.9 | 19.4 | 19.3 | 16.1 | 19.4 |
| Mica and chlorite | 11.4 | - | - | 9.3 | 7 | 9.4 | 6.6 | 9.1 |
| Clay minerals | 4.3 | - | - | 1.6 | 1.4 | 1 | 0.9 | 1.8 |
| Minor silicates | 0.5 | 0.1 | - | 2.9 | 2.4 | 2.3 | 1.6 | 2.3 |
| Carbonates (Ca, Mg, Fe) | 21.1 | 0.1 | 5.0 [1] | 19.6 | 15.2 | 19 | 17.1 | 18.5 |
| Gypsum | 9.7 | - | 2.9 | 5.2 | 5.7 | 5.1 | 3.5 | 6.4 |
| Ca-phosphate | 0.5 | - | - | 0 | 0 | 0 | 0 | 0 |
| Ettringite | - | - | 0.8 | 12.2 | 17.5 | 10.1 | 29.5 | 12.1 |
| Ca-Al-Mg-silicate (slag) | - | 99.1 | 20.7 | 6.3 | 5.3 | 10.3 | 1.9 | 5.9 |
| Metal oxides and hydroxides | 10.3 | 0.6 | 3.1 | 0.9 | 0.9 | 0.8 | 0.9 | 0.6 |
| Metal sulphides | 0.4 | - | - | 0.1 | 0.2 | 0.2 | 0.2 | 0.5 |
| Ca-silicates (C$_2$S, C$_3$S) [2] | - | - | 53.2 | - | - | - | - | - |
| Ca-aluminate (C3A) [2] | - | - | 9.3 | - | - | - | - | - |
| Ca-Al-Fe-oxide (C4AF) [2] | - | - | 4.7 | - | - | - | - | - |
| Unclassified | 0.7 | 0.1 | 0.1 | 0.6 | 0.5 | 0.7 | 0.4 | 0.6 |
| Total | 100 | 100 | 100 | 100 | 100 | 100 | 100 | 100 |
| Matrix (<10 µm), not included in results | - | - | - | 25.7 | 28.8 | 21.3 | 25.2 | 26.2 |

[1] Calcite. [2] Cement clinker phases.

According to the sedigraph measurements, mine tailings grain size was <500 µm and 90% of the volume distribution was below 82 µm, 50% below 10 µm and 10% below 1.7 µm. Corresponding values measured by MLA for slag and cement are: grain-size <75 and <65 µm, 90% below 35 and 30 µm, 50% below 12 and 14 µm and 10% below 6 and 6 µm, respectively. The small particle-sizes of starting materials resulted in the formation of very fine-grained (<5 µm) intergranular matrices in CPB specimens, which tend to produce mixed EDX spectra and hamper the MLA analyses. To avoid these drawbacks, we collected several areal (25–100 µm$^2$) EDX analyses from the matrices of each specimen and used them to count out the mixed spectra in the results table. The data for CPB specimens shown in Table 6 thus represent grains, which are roughly more than 5 µm in diameter, whereas the smaller particles are not included in the results. It should be noted that, although the CPB specimens were prepared from starting materials that are very similar to those examined in this work, they are from different batches and therefore do not directly represent comparable compositions.

As seen in Table 6, the phase compositions of CPB are dominated by the mine tailings, with an addition of abundant ettringite (10–30 wt.%). These are interpreted from EDX spectra and back-scatter electron (BSE) images (see Figure 1C,D) to be composed primarily of hydrated reaction products between starting materials and water, although the lightest elements such as hydrogen inter alia cannot be directly measured by SEM EDX. The interpretation is supported by Raman microscopy, which indicates that the acicular phase, for example, seen in Figure 1C, is ettringite [Ca$_6$Al$_2$(SO$_4$)$_3$(OH)$_{12}$], one of the most common reaction products between tricalcium aluminate (C$_3$A) and gypsum (CaSO$_4$·2H$_2$O) in

cement. The gypsum content of the mine tailings (10 wt.%) and cement (3 wt.% observed, 5 wt.% stated by the manufacturer) was reduced in the CPB samples (4–6 wt.% observed). This is partly explained by dilution effect resulting from the addition of the binders and partly by the consumption of gypsum in the curing reactions. As predicted, the release of $SO_4^{2-}$ resulted in development of ettringite (10–30 wt.% observed) in CPB. The rapid early formation of abundant ettringite would also consume an excess of free water in the samples. When the cement is the only binder (sample R4), the secondary gypsum is most efficiently consumed: only 4 wt.% remaining after 13 wt.% (in addition to sulphate in water) was observed in the starting materials. The corresponding high content of observed ettringite (30 wt.% in R4) together with the remaining excess gypsum indicates that most free aluminate and iron were consumed by the formation of ettringite.

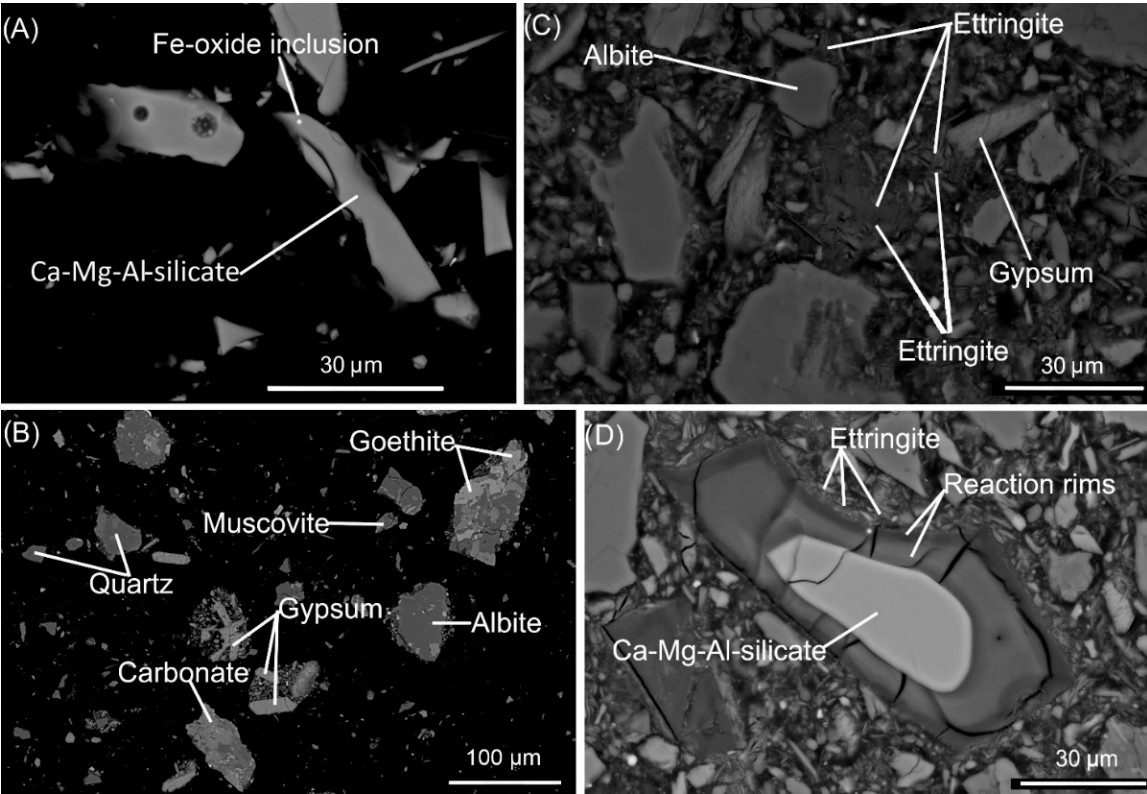

**Figure 1.** Back-scattered electron (BSE) images of the studied materials. (**A**) Slag sample (Ca-Mg-Al-silicate), with a characteristic Fe-oxide inclusion. (**B**) Tailings sample showing lithic fragments composed mainly of quartz, albite, and carbonate. Some liberated gypsum and muscovite grains are also shown in the image. (**C**) Sample R4, with ettringite filling spaces in the matrix and grows along the boundaries of other minerals such as albite (marked in the image). (**D**) Sample R3. Grain of Ca-Mg-Al-silicate (slag), with an unreacted core (bright) and reaction rims (darker grey) that are obviously hydrated (notice abundant drying cracks, and an evaporated hole after the electron beam on the lower right-hand side).

The minor reduction of carbonates and phyllosilicates (clay, chlorite, mica) of all CPB specimens, in comparison with mine tailings, is partly explained by the dilution effect resulting from the addition of binders although the content in R2 and R5 was over 2% lower compared to other specimens, which suggests Al consumption in ettringite formation.

The other compositional endmember of the CPB test sample with all the cement replaced with slag (R3) showed comparatively high remaining gypsum content (5.1 wt.%) and the lowest retained ettringite content (10.1 wt.%), whereas mixtures of cement and slag binders provided intermediate gypsum and ettringite contents between those of R3 and R4. This indicates that the replacement of cement with slag inhibited the formation of ettringite. Slags are known to produce similar phases to those that are formed during the hydration of cement when mixed with water. However, slag needs an activator for this to happen,

for example cement, $Na_2O$, $K_2O$ or $CaSO_4{}^{2-}$ [35]. The mine tailings were neutralized with hydrated lime after the enrichment process, which resulted in the precipitation of gypsum [36,37]. The activation mechanism of slag here was assumed to be a reaction enabled by gypsum and slag-derived $Ca^{2+}$ and $Al(OH)_4{}^{-}$ ions [9], which resulted in the early precipitation of ettringite. Excess dissolved calcium would enable the precipitation of calcium hydroxide (portlandite), and silica released during the dissolution should form a C–S–H gel [35]. No pure Ca-oxyhydroxide or Ca-silicate hydrate phases were observed in the CPB test samples by the MLA. Our interpretation is that no early excess Ca was present to form portlandite and that the signal from the amorphous C–S–H gel was likely to be integrated into the intergranular matrix that was not included in the classification.

The slag of the starting material was relatively homogeneous in chemical composition and formed clear grains for the MLA to detect. The highest quantity of slag in the CPB samples was observed in R3, which had no cement added and the observed slag content of over 10 wt.% clearly exceeded the 7 wt.% added from the CPB recipe. This phenomenon resulted from the hydration reactions of the slag, which are clearly visible as reaction rims around some unreacted slag cores (Figure 1D). The hydration increased the volume of the slag particles and reduced their average density, both of which reflected an increase in slag content and also a potential overestimation of the slag content as determined by MLA. The EDX observation (Figure 2) was that the altered rims were depleted in Ca, Al, and Si relative to the fresh slag glass. We suggest this related to the formation of ettringite along the grain boundaries of hydrated slag grains (Figure 1D), which in this case would consume Ca from both gypsum and hydrated slag in addition to Al, and possibly Si [38], from the slag.

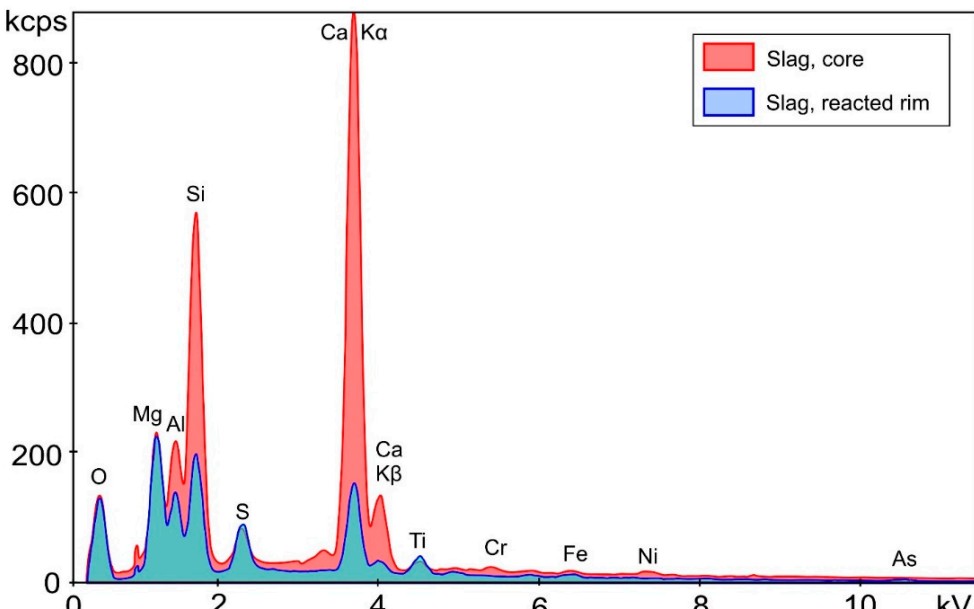

**Figure 2.** Energy-dispersive X-ray spectroscopy (EDX) spectrum of the unreacted core of Ca-Mg-Al-silicate (slag) and the EDX spectrum of the reaction rim. Based on these, data suggest the reaction rim is depleted in Ca, Al, and Si in comparison with the core, which presumably relates to growth of ettringite on the grain boundary (see text for more details).

### 3.3. Compressive Strength

Most of the CPB mixtures with the binder addition rate of 7% achieved the compressive strength of 0.5 MPa within the 14-day curing period required by the mine for CPB (Figure 3). If the binder was included at only 5%, the compressive strength of 0.5 MPa should have been achieved in 28 days. The recipes R1 (40:60), R2 (50:50), R4 (100:0) and R5 (50:50) exceeded the required 0.5 MPa in the compressive strength tests after 14 days of curing, although R5 (50:50) had only 5% of binder. Recipes R1 and R2 containing mixtures of slag

and cement had very good compressive strength results exceeding 1 MPa by 14 days. The sample with only slag binder (R3) did not reach the required strength by 14 days, but after 56 days of curing the strength had already reached as high as 1.3 MPa, which is comparable to CPB with pure cement binder (R4).

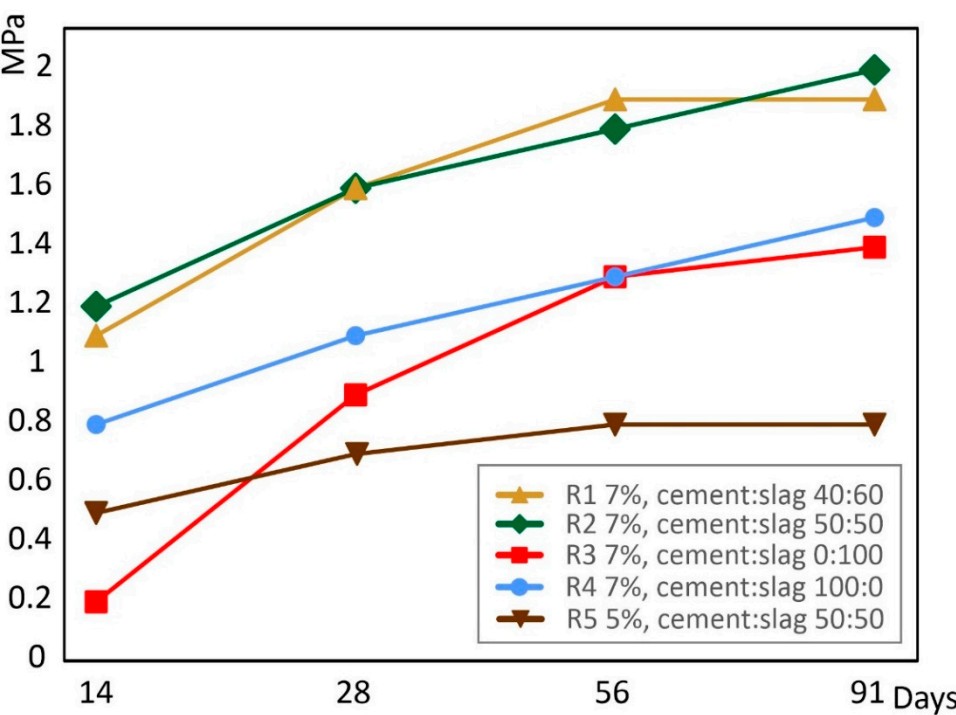

**Figure 3.** Compressive strength development of the CPB samples with different binder compositions (R1–R5, see Table 1.). A strength of 0.5 MPa should be achieved or exceeded within 14 days of curing for the specimens that have 7% of binder material (R1–R4) and in 28 days for a specimen containing 5% of binder material (R5). Only R3 did not have the desired strength at 14 days, but after 56 days of curing the R3 strength value was identical with that of R4.

Slag needs an activator to activate its latent hydraulic properties [28]. The relatively high compressive strength obtained for the sample containing only slag as a binder material (R3) implies that pore water and/or mine tailings used for the CPB specimens preparation contained substances for slag activation. When the activator is not cement, it is most likely to be the gypsum that is formed during the neutralization of mine tailings and process water by hydrated lime. Activation of slag with gypsum forms primary ettringite and a reaction between hydrated lime of cement and slag forms additional C–S–H gel [39]. Both reactions enhance the compressive strength results. These short-term investigations suggest that recipes in which part of the cement is replaced with slag are suitable for CPB materials. In addition, total replacement of cement with slag in CPB would be applicable, if rapid hardening is not required. However, it must be considered that long-term tests are still to be executed and the quality on mine tailings and water are variable. Water and tailings quality changes should be carefully monitored if slag only was to be used as a binder material for CPB.

### 3.4. X-ray Tomography

X-ray tomography was used to observe the potential porosity, cracking and other structural changes of the CPB test specimens. The samples were cured for 65–67 days before they were imaged with X-ray tomography.

All samples were shown to be denser at the bottom of the sample compared to the upper half (Figure 4). The samples were least dense at the top surface, followed by a slightly denser layer peaking at the depth of ~15 mm, followed by a less dense layer and finally a

steep gradient towards more dense material at the bottom. The observed grayscale changes were quite small, a few percent at most. A possible cause for this phenomenon is likely to be connected to the compaction of solid particles in the base and upward injection of the liquid phase. This type of process is thus likely to occur in other cases of CPB test specimen preparation and warrants more research. In contrast, the dense layer at the top may be related to sample preparation. There is thus a possibility that similar density changes appear in the CPB mass in a mine cavity, although this will depend on the placement method and sequence used. It is not clear what would be the consequences of this type of variation in density in different parts of the mass and there is a need to study this topic further.

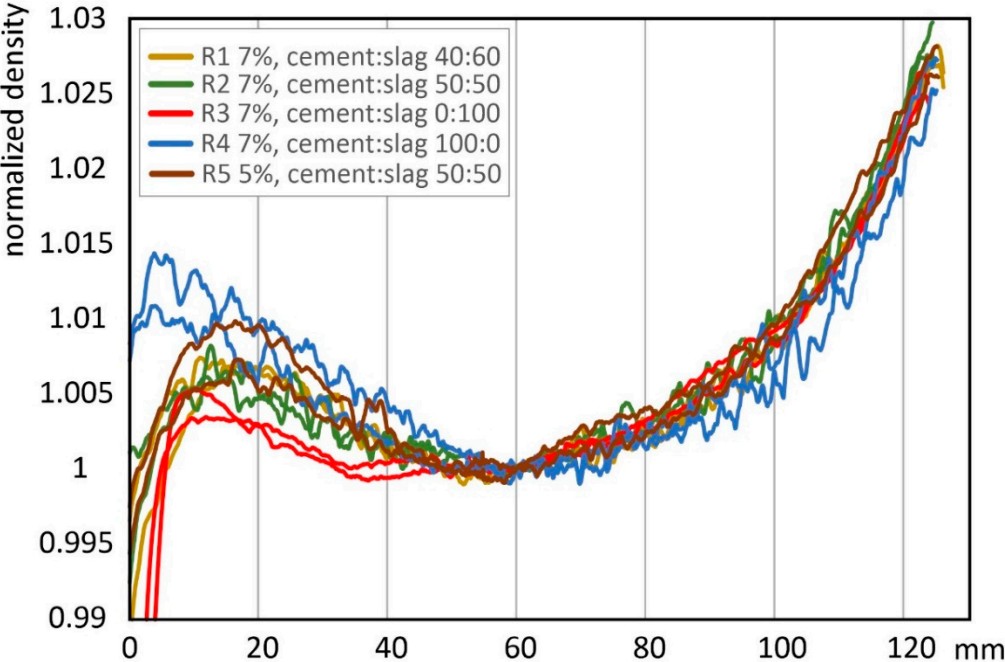

**Figure 4.** Average density / grayscale value of each XY-plane of each sample normalized by dividing by the density at z = 60 mm, showing similar trends for all samples. CPB specimens' top is on the left. CPB sample (R1–R5) compositions are presented in Table 1.

The porosity results obtained using X-ray tomography had two clear standouts. The sample with pure slag binder (R3) was by far the least porous with 0.051% total porosity, and the sample with pure cement binder (R4) was the most porous with 0.52% (Figure 5). The samples with cement/slag mixtures were in the range of 0.22–0.32% (Table 7). No cracking was observed.

**Table 7.** Porosities of all CPB samples measured in X-ray computed tomography (XCT) images.

|  | CPB Specimens (Cement:Slag Ratio) | | | | |
| --- | --- | --- | --- | --- | --- |
|  | R1 (40:60) | R2 (50:50) | R3 (0:100) | R4 (100:0) | R5 (50:50) |
| Sample 1 | 0.22% | 0.31% | 0.06% | 0.56% | 0.21% |
| Sample 2 | 0.21% | 0.32% | 0.04% | 0.48% | 0.22% |
| Average | 0.22% | 0.32% | 0.05% | 0.52% | 0.22% |

Slag is known to produce a dense microstructure and good resistance against chemical attack because of slow reactivity [35]. In slag-containing samples the hydration time has been longer and degrees of released heat lower, which has resulted in lower porosity especially for the sample where only slag binder was used (R3). The higher porosity of cement-containing samples is probably caused by the reactive calcite content and the rapid curing.

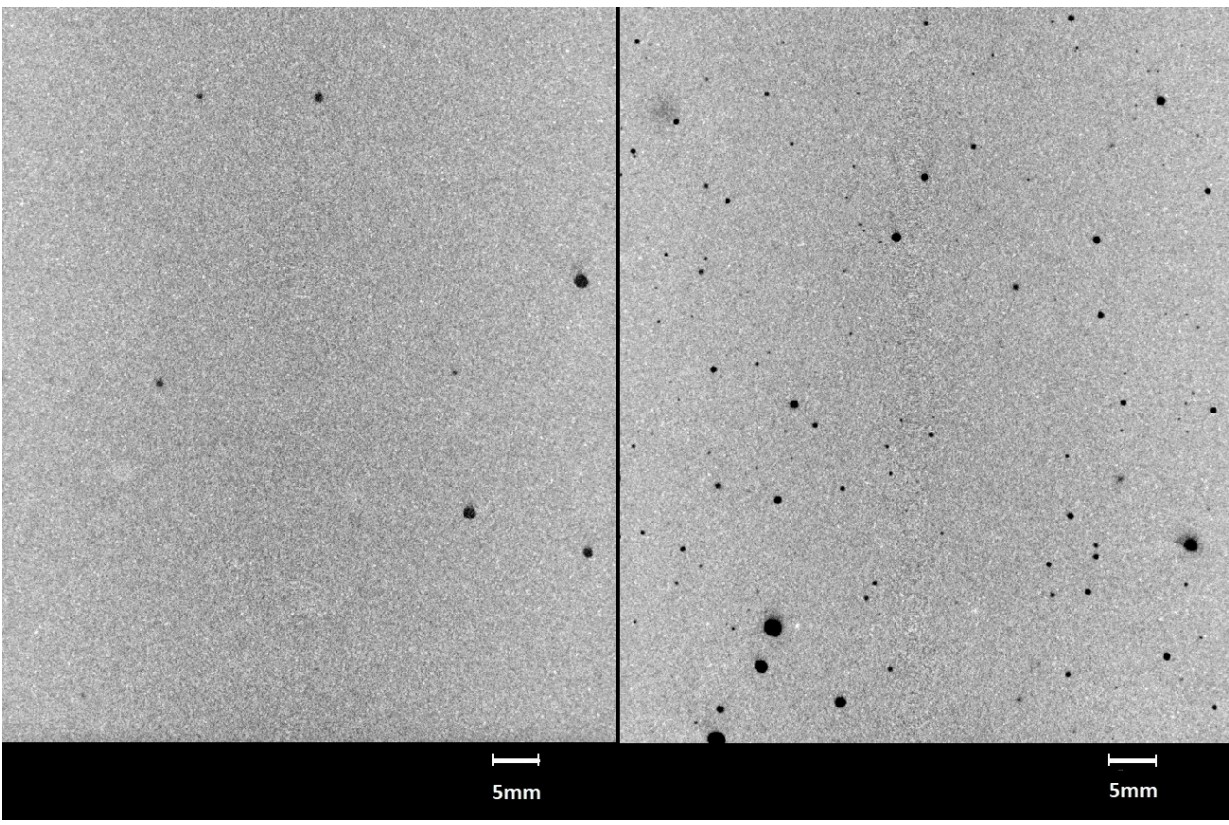

**Figure 5.** Porosity differences between the samples R3 (0:100, cement: slag left) and R4 (100:0, cement: slag right).

## 4. Conclusions

We studied five CPB test recipes (R1–R5) by varying their binder inclusion quantities and their cement–slag ratios. The lime-washed mine tailings in this study are rich in inert silicates, slightly reactive dolomite-ankeritic carbonates and reactive secondary gypsum. The Portland cement used contained slag, calcite and gypsum, and the slag was predominantly Ca-Fe-Mg-Al-silicate glass containing 6 wt.% aluminium (11 wt.% $Al_2O_3$). The properties of each recipe were determined by bulk geochemistry of the CPB specimens and solid and liquid starting materials with four acid, aqua regia and two-step batch leaching tests. CPB test samples quantitative phase composition was measured with MLA, in addition to compressive strength tests and internal structure observation with the XCT method.

The main environmental issues raised were the As and $SO_4^{2-}$ leachability of CPB specimens. Mixing alkaline binder materials with mine tailings raises the pH and thus increases As leachability in distilled water for all CPB recipes. The excess $SO_4^{2-}$ content in all CPB specimens creates a possibility for sulphate attack. Further study is in preparation to clarify how As and $SO_4^{2-}$ of a CPB specimen reacts in contact with real mine drainage water instead of distilled water. If CPB materials were to be exposed to severe AMD, the concern would be, in addition to leaching of As and $S^{2-}$, potentially Cu, and Ni, Sb and Co, but this is unlikely to happen in this case, since the tailings are low in sulphide minerals.

CPB specimen structure was revealed to be heterogenous. Density was slightly higher on the bottom of all five samples. This effect can be a result of compaction of tailings particles in the base and upward injection of the liquid phase. More research is needed to find out why this phenomenon occurs and its consequences.

The following conclusions were drawn concerning the differences between CPB recipes structural and phase composition properties:

The test samples with mixed cement/slag binder (R1, R2 and R5) seem to be suitable for CPB materials based on compressive strength tests. Recipes containing 7% binders

with the cement slag ratios 40:60 and 50:50 (R1 and R2) achieved over 1 MPa compressive strength results within 14 days of curing, which was much higher in comparison with other recipes. Partial replacement of cement with slag improved the compressive strength when 7% of binder was used. One explanation for the very good compressive strength properties of R1 and R2, could be the C–S–H formation that probably resulted from the reactions between hydrated lime of cement and slag. The specimen in which the cement slag ratio was 50:50 and total binder proportion was 5% (R5) achieved 0.5 MPa limit value in 14 days, which is lower but sufficient for backfilling material. However, the leaching properties of As indicates that 5% is not suitable amount of binder for this type of tailings.

When only slag was used as a binder R3, the early strength requirement was not achieved. However, R3 showed potential for applications for which rapid curing is not required, because the 56th day strength for R3 was the same as for the sample containing only cement as the binder material (R4). R3 achieved the lowest porosity of all the CPB recipes. The explanation for this lies in long hydration time and low heat of hydration compared to other cement-containing samples. We assume that the activator for R3 was gypsum, which is formed in the reaction between hydrated lime and $SO_4^{2-}$ rich parent materials. Replacement of cement with slag inhibits the formation of ettringite when compared to higher cement content samples but it seems that the excess calcium and silica released during hydration reactions would have formed C–S–H gel, which would explain the relatively high strength properties along with low porosity.

The CPB test samples with cement-only binder (R4) showed fast early strength development, but intermediate final strength compared to R1 and R2. Tomography results showed the greatest extent of porosity for cement-based CPB, which we interpreted as resulting from the reactive calcite and rapid curing. The very high ettringite content observed is consistent with a high total gypsum content of the tailings and consumption of Al and Fe that is evidenced by the decrease in mica/clay component. Ettringite is probably the main cohesive phase in this cement only sample, which would explain the intermediate final strength.

**Author Contributions:** S.S.: Conceptualization, methodology, Investigation, Writing—Original Draft, Project administration. A.T.: Investigation, Funding acquisition, Writing–review and editing. P.H.: Investigation, visualization, Writing—review and editing J.K.: investigation, Writing—review and editing. S.H.: Methodology, Investigation, Writing—review and editing. P.J.: Writing—review and editing, Project administration. M.B.: Writing—review and editing. T.K.: Conceptualization, Supervision, Funding acquisition, methodology, Writing—review and editing. All authors have read and agreed to the published version of the manuscript.

**Funding:** This research is supported by European Regional Development Fund (grant number: A75014, Funding date: 1 June 2019–28 February 2022), Geological Survey of Finland and business partners Agnico Eagle Finland Ltd., Kuopion Energia Ltd., Finnsementti Ltd., Yara Suomi Ltd., Normet UK Ltd., and Endomines Ltd.

**Acknowledgments:** The authors would like to thank the Kove-Pro steering group and all project members for co-operation and contribution. In addition, we would like to thank Alisdair McLean for proofreading the manuscript and all reviewers and editors whose comments improved the manuscript significantly.

**Conflicts of Interest:** The authors declare no conflict of interest. The involvement of the industry is a prerequisite of ERDF funding.

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
