# Peer review of "Substitution of Cement with Granulated Blast Furnace Slag in Cemented Paste Backfill: Evaluation of Technical and Chemical Properties"

_minerals, doi:10.3390/min11101068_

Round 1

Reviewer 1 Report

This study This work focuses on the technical and chemical 
evaluation of the incorporation of Granulated Blast Furnace Slag in Cemented Paste Backfill. This paper is interesting and well written.

Based on the quality of the manuscript, the review thinks that the paper should be accepted

Author Response

Dear Reviewer,

Thank you for reviewing the paper and for the positive comments.

Minor spell check has been done; I observed and corrected some typos which happened when I edited the article to fit the style of the Minerals. The article had been proofread by a professional before that. Other reviewers found a couple of issues concerning the language. Those issues are now corrected and hopefully the language quality is at an adequate level now.

Sincerely,

Soili Solismaa

Reviewer 2 Report

Journal Minerals (ISSN 2075-163X)

Manuscript ID minerals-1389137

Type: Article

Number of Pages: 20

Title: “Substitution of Cement with Granulated Blast Furnace Slag in Cemented Paste Backfill: Evaluation of Technical and Chemical Properties”

Authors: Soili Solismaa, Akseli Torppa, Jukka Kuva, Pasi Heikkilä, Simo Hyvönen, Petri Juntunen, Mostafa Benzaazoua, Tommi Kauppila

General comments:

Although it is not a novel topic, I think that both the title of the paper and the research done by the authors may be interesting for the readers. The authors have made a thorough description of the materials and methods, and the discussion is solid and convincing. The conclusions are coherent and in harmony with the development of the research. It is a work that deserves to be published in Minerals.

However, some comments are made to the authors. See below:

Abstract: Authors have to rewrite the Abstract as follows: First, a brief presentation of the subject of the work; second: a brief presentation of the methods used in the research; third: a quick description of the results obtained; fourth: define in which field the results of this research will be applied. Review.

Line 78. Remove the comma before the reference number.

Lines 82 to 89. Move this paragraph to Section 2 "Materials and Methods".

Section 2. "Materials and Methods". Authors should divide this section into two subsections: 2.1 "Materials"; 2.2 "Methods". Review.

Section 2.1. "Mine site and Sampling Description". It is too long. Should be shortened. Revise.

Author Response

Dear Reviewer,

Thank you for reviewing the paper and for the valuable instructions to improve the paper quality. I have made changes according your suggestions:

Point 1:

Abstract: Authors have to rewrite the Abstract as follows: First, a brief presentation of the subject of the work; second: a brief presentation of the methods used in the research; third: a quick description of the results obtained; fourth: define in which field the results of this research will be applied. Review.

Response 1:

Abstract is now revised according your advice:

Abstract: Cemented paste backfill (CPB) offers an environmentally sustainable way to utilize mine tailings, one of the largest waste streams in the world. CPB is a support and filler material used in underground mine cavities, which consists of mine tailings, water, and binder material that usually is cement. Replacing cement with secondary raw materials like granulated blast-furnace slag reduces the total CO2 emissions and strengthens the internal microstructure of the CPB.

This study characterizes the total- and soluble contents of CPB starting materials and five CPB specimens containing different levels of slag substitution. In addition, phase composition (mineral liberation analysis, MLA) and internal structure (X-ray tomography) of five CPB specimens is documented, and measurements of compressive strength are used to evaluate their suitability as backfill material.

Mine tailings and CPB specimens used in this study are rich in sulphates and arsenic, but low in sulphides. Stronger As leaching of ground CPB specimens compared with ground mine tailings is related to the elevating pore water pH resulting from the cement hydration. The hydration product ettringite is found in all CPB specimens and its content is the lowest in the slag containing specimens. X-ray tomography revealed vertically differentiated density structures in the CPB specimens. The lower parts of all specimens are denser in comparison with the upper parts, which is probably due to the compaction of the solid particles at the base. The compressive strength test results indicate that partial substitution of cement with slag improves the strength of the CPB. The total replacement of cement with slag reduces the early strength but gives excellent strength and lower porosity over longer time intervals.

The results of the study can be utilized in developing more durable and environmentally responsible CPB recipes for gold mines of similar mineral composition and gold extraction method.

Point 2:

Line 78. Remove the comma before the reference number.

Lines 82 to 89. Move this paragraph to Section 2 "Materials and Methods".

Response 2:

Grammar correction is done and the paragraph moved to section 2.

Point 3:

Section 2. "Materials and Methods". Authors should divide this section into two subsections: 2.1 "Materials"; 2.2 "Methods". Review

Response 3:

I agree. “Mine site and sampling” is now “Materials” and “Chemical, mineralogical, and secondary phase composition determination of parent materials and CPB recipes” has now “Methods” as a headline.

Point 4:

Section 2.1. "Mine site and Sampling Description". It is too long. Should be shortened. Revise

Response 4:

Now, when Section 2.1. is headlined “Materials” it is hopefully not too long. As a geologist I really want to say something about the geology of the ore. The ore enrichment process description is needed, since there are two types of tailings formed in the mine site and the enrichment process effects very much on the tailings properties. The description concerning the process water used in the study is shortened.

Sincerely,

Soili Solismaa

Reviewer 3 Report

Very well written paper about several aspects of CPB based on tailings and cement/GGBFS binder. I have just a few minor comments and recommendations.

Line 59: C-S-H, not C-H-S.

Line 361: The clinker minerals are usually written as C3S (not C3S).

Line 335: "metal oxides and hydroxides" - which ones?

Line 427: "hydraulic lime" is probably not correct. I suppose that tailings are washed by suspension of "common" lime - Ca(OH)2 (hydrated lime, slack lime). Real "hydraulic lime" is rather unusual material containing CaO, C2S etc.

Line 444, Fig 2: I'm not sure if such EDX comparison is correct

Author Response

Dear Reviewer,

Thankyou for reviewing the paper and giving valuable comments to improve the paper. I have made changes to the manuscript according your suggestions:

Point 1

Line 59: C-S-H, not C-H-S.

Line 361: The clinker minerals are usually written as C3S (not C3S).

Response 1:

These are now corrected. I found two other places where I had been used C-H-S instead of C-S-H, they are also corrected now.

Point 2

Line 335: "metal oxides and hydroxides" - which ones?

Response 2:

Here probably the line number 335 means line 365, since metal oxides and hydroxides are not mentioned in the line 335. The metal oxides and hydroxides are comprised oxides and hydroxides of iron. They are presented together in the text and in the table 6 since it was not possible to divide them to separate phases.

Point 3

Line 427: "hydraulic lime" is probably not correct. I suppose that tailings are washed by suspension of "common" lime - Ca(OH)(hydrated lime, slack lime). Real "hydraulic lime" is rather unusual material containing CaO, C2S etc.

Response 3:

Thankyou very much for noticing this, hydrated lime is the right term, and it is now corrected.

Point 4

Line 444, Fig 2: I’m not sure if such EDX comparison is correct

Response 4:

We find the comparison valid. EDX measurements were measured sequentially from the different zones of the same grain. Similar kV/mA adjustment and beam diameter were used in both measurements. Count rate scale is the same for both spectrums. Figure demonstrates the changes in composition between the hydrated part and unreacted slag

Sincerely,

Soili Solismaa